# Community perception of abortion, women who abort and abortifacients in Kisumu and Nairobi counties, Kenya

**Boniface Ayanbekongshie Ushie**[☯]**, Kenneth Juma**[iD]**\*[☯], Grace Kimemia**[☯]**, Ramatou Ouedraogo**[‡]**, Martin Bangha**[‡]**, Michael Mutua**[‡]

African Population and Health Research Center, Nairobi, Kenya

☯ These authors contributed equally to this work.
‡ These authors also contributed equally to this work.
\* kjuma@aphrc.org

## Abstract

### Background

Abortion draws varied emotions based on individual and societal beliefs. Often, women known to have sought or those seeking abortion services experience stigma and social exclusion within their communities. Understanding community perception of abortion is critical in informing the design and delivery of interventions that reduce the gaps in access to safe abortion for women.

### Objective

We explored community perceptions and beliefs relating to abortion, clients of abortion services, and abortifacients in Kenya.

### Methods

We conducted focus group discussions (FGDs) and in-depth interviews (IDIs) in Kisumu and Nairobi counties in Kenya among a mix of adult men and women, pharmacists, nurses, and community health volunteers.

### Results

Community perspectives around abortion were heterogeneous, reflecting a myriad of opinions ranging from total anti-abortion to more pro-choice positions, and with rural-urban differences. Notably, negative views on abortion became more nuanced and tempered, especially among young women in urban areas, as details of factors that motivate women to seek abortion became apparent. Participants were mostly aware of the pathways through which women and girls access abortion services. Whereas abortion is commonplace, multiple structural and socioeconomic barriers, as well as stigma, are prevalent, thus impeding access to safe and quality services.

**Data Availability Statement:** The data underlying the results presented in the study are available from (APHRC, VIA the info@aphrc.org).

**Funding:** This work was supported by the Hewlett and Packard Foundation under Grant number 2015: 3063 to the African Population and Health Research Center (APHRC), https://hewlett.org/. The funder had no role in study design, data collection and analysis, decision to publish, or preparation of the manuscript.

**Competing interests:** The authors have declared that no competing interests exist.

## Conclusion

Community perceptions on abortion are heterogeneous, varying by gender, occupation, level of education, residence, and position in society. Stigma and the hostile abortion environment limit access to safe abortion services, with several negative consequences. There is urgent need to strengthen community-based approaches to mitigate predisposing and enabling factors for unsafe abortions.

## Introduction

The rising burden of unsafe abortion and the resultant magnitude of severe complications constitute a significant public health challenge in sub-Saharan Africa (SSA) [1]. The high incidence of unsafe abortion belies the growing availability of safer and quality procedures, based on the WHO guidelines, for terminating pregnancies [2]. Across SSA, unsafe abortion remains prevalent, accounting for up to 29% of the global burden of unsafe abortions, and about 62% of the 2014 global abortion-related deaths [3]. While fertility is highly desirable in SSA and infertility prevalent [4], abortion draws strong objections, underpinned by religious, moral, ethical, socio-cultural, and medical concerns, and remains highly disapproved. Cultural and religious intolerance to abortion, among communities and service providers–manifesting most saliently as abortion stigma, as well as the cost of care–continues to drive women and adolescent girls to self-managed abortion procedures or those offered clandestinely mainly by unqualified providers.

Existing evidence in SSA establishes high levels of severe maternal outcomes attributable to adverse complications such as sepsis, cervical and uterine ruptures, hemorrhage, and death [5, 6]. Strong anti-abortion laws have not translated into a reduction in the incidence of abortion; instead, they have increased the magnitude of unsafe abortion [7]. In Kenya, for example, annual abortion estimates increased from 300,000 in 2004 [8] to 445,000 in 2012 [9], despite the environment remaining highly restrictive through laws that criminalize abortion. Nevertheless, while several countries across the globe have revised, or are reviewing their abortion laws to embrace more liberal and pro-choice principles [10], criminalization remains predominant in Africa.

Article 26 of the 2010 Constitution of Kenya widened the latitude under which abortion is permissible, including when "in the opinion of a trained health professional, there is a need for emergency treatment, or the life or health of the mother is in danger, or if permitted by any other written law" [11]. However, older laws and statutes that criminalized abortion remain unrepealed and continue to form basis for judgements on abortion litigations in the country. Subsequently, the Ministry of Health issued, and later withdrew, standards and guidelines on safe abortion care [12], which created confusion among care providers and the public, with the withdrawal interpreted as a complete restriction on all abortion services. This ambiguity fuelled increased public anti-abortion stance and stigmatization, implying that other than addressing legal restrictions, changing cultural beliefs and attitudes towards abortion at the community level will be necessary levers for the reduction in unsafe abortion[13].

Since 2015 the WHO added misoprostol and mifepristone–medical abortion drugs–to the list of essential medicines [14, 15], but in SSA, their availability, accessibility, and acceptability is severely limited. Questions remain about the potential impact on the incidence of unsafe abortion of sufficiently stocking misoprostol and mifepristone at access points in the communities and distributing to clients with proper guidance by pharmacists at affordable prices [16].

Although legal restrictions currently limit access to medical abortion products, existing evidence shows that an increasingly large proportion of women are dependent on them,

especially at the community level within private clinics and pharmacies, when backed with a prescription [16–18]. Young unmarried women and girls, prefer using access points like pharmacies and chemists because they provide greater anonymity compared to hospitals and clinics [19] and shields the users from stigmatization.

Transforming community perspectives toward abortion can shape access to safe abortion as well as post-abortion care practices [20]. The first step to changing people's perception is to capture the prevailing views within the targeted communities to inform the design of community interventions that address gaps in access to safe abortion for women. In this paper, we explored community-level perceptions of abortion and examined the various modes of access and use of abortifacient pharmaceutical products in Kisumu and Nairobi counties in Kenya. The significance of this community-level study is the inclusion of men–who are mostly responsible for decision-making at the household level, even on issues that involve women's health–and care providers such as pharmacists, community health volunteers (CHVs), medical officers (MOs) and clinical officers (COs).

## Methods

We conducted a cross-sectional exploratory community-based study to understand community-level perceptions of abortion and to explore access and use of abortifacient pharmaceutical drugs in Kisumu and Nairobi Counties in Kenya between May and October 2017. Within these two counties, communities representing rural, peri-urban, urban informal and urban formal settings were included.

Men and women of reproductive age (15–49) within the communities were organized into groups of 8–12 members for the Focus Group Discussions (FGDs). In-depth Interviews (IDIs) were conducted with critical reproductive health experts (including nurses, pharmacists/pharmaceutical technologists, CHVs/community health extension workers (CHEWs), as well as other knowledgeable and relevant community members such as teachers, women thought-leaders, sex workers, bar/pub operators and beauty salons attendants among others.

Trained interviewers conducted the IDIs and FGDs with the identified participants. The IDIs solicited information on unique perspectives of individual respondents, while the FGDs generated information on community-level knowledge and norms about abortion. Overall, we conducted a total of 57 IDIs (including 36 women and 12 men; as well as pharmacists (2), CHVs (3), and nurses (4))–and 18 FGDs—nine each for men and women. An additional fifteen (15) key informant interviews were conducted with clinical officers, health records officers and laboratory technologists. The interview guides covered issues ranging from unintended pregnancies, abortion, community attitudes toward women who have abortions, available drugs or medicines for abortion, methods used to induce abortion, perceived need for information on abortion methods and sexual and reproductive health, and preferred sources of such information in the community. The guides also covered issues such as abortion providers targeted for use by women, and social and cultural norms that influence access to safe abortion services. We collected information to help us provide broad descriptions of respondents' personal, religious and cultural values regarding abortion and how it shapes their perspective on abortion services and women seeking these services.

The study protocol, including the interview guides, was approved by the Kenya Medical Research Institute (KEMRI) Scientific and Ethics Review Unit (SERU) on December 12, 2016 (protocol ID: KEMRI/RES/7/3/1). All participants consented to participate in the study by providing a signed or thumb-printed informed consent form. We translated all the interview guides to Kiswahili and Dholuo languages, which are the most widely used in the target counties.

Interviews were recorded digitally, transcribed verbatim, and translated to English (for those in Kiswahili or Dholuo) by an expert translator. All transcripts including field notes (records of discussions, observations and non-verbal cues, and the context of the interviews) were uploaded into NVivo version 10 (QSR International) for coding by the researchers.

Data analysis was performed using the thematic framework approach, as described by Ritchie and Spencer [21]. Authors reviewed the same set of transcripts using a phenomenological approach to develop codes and code definitions, and where conflicts arose, they were resolved by consensus. After reviewing transcripts for FGDs and IDIs, we consolidated identified codes and summarized them into a codebook. The emerging codes were applied to relevant blocks of text in the remaining transcripts in NVivo. Whenever new codes emerged in the data, we added them to the codebook. Codes were summarized in analytical memos and illustrated using direct quotes from the participants. All authors reviewed the codebook and memos and discussed emergent themes. Conversations about the findings among the authors allowed for a discussion of alternative explanations, a better understanding of the local context, and opportunities to address issues of reflexivity. Key themes identified included: perceptions of abortion, awareness of, and access to medical abortion services.

## Results

### Perceptions of abortion

**Abortion is a synonym for deviance and ostracization.** Participants recognized and acknowledged abortion as commonplace in the community. However, the dominant view expressed by both female and male participants is that abortion is objectionable and a deviation from the social norms and perspectives embedded in religious and cultural beliefs. Considering the negative community opinions related to pregnancy termination, and the fear of shame and stigma resulting from it, abortion is often conducted in secrecy, and women disclose this information, if at all, only to their close friends and confidantes. Women who are exposed as having had an abortion are ostracized, labeled and stigmatized as murderers and prostitutes. They are often perceived as bad examples to younger women and often excluded from community activities. This is described by a teacher in an urban slum in Nairobi and a young woman in rural Kisumu thus:

*She's like a black sheep in society. She is usually condemned. She is like a bad example to the rest of the girls, so people don't like associating with such women. Yes, she is isolated* (IDI, Male Teacher, 34 years, Jericho, urban setting).

*Most people will view them as people who like having abortion now and then, so they are not viewed in good light. They are not involved in various community activities* (IDI, Young unmarried woman, 19 years, Kisumu, rural settings).

Ostracization of women may be individual-level guilt, but the community may also exert negative pressure that can lead to adverse outcomes such as poor mental health and suicide, as detailed in a discussion among older married men in an urban slum.

*In most cases, it brings shame to you in the community because whenever you walk out, people discuss you. In most cases, you isolate yourself because personally, you have self-suspicions* (FGD, Married men, 37 years, Viwandani, urban setting).

Discussions with community members revealed the extent to which women known to have terminated their pregnancies face extreme stigma and are isolated and shunned. Some women

resort to abandoning their livelihoods, temporarily or permanently relocating to a different community. Women who relocate temporarily believe that given enough time people will forget about the abortion story and move on with their lives. Due to the pain of social isolation, women tend to find new residences where no one knows their past to facilitate their integration into the community, as reported by young men in Kisumu.

*First, if someone secures an abortion, they may leave that community for some months because of the embarrassment because people knew she was pregnant, and she is no longer pregnant. . ..Mostly they don't stay with people in their community when they do that because of the embarrassment* (FGD, Young men, 22 years, Nyalenda, urban setting).

However, relocation negatively impacts the individuals, disrupting their livelihoods and their families. For younger girls, this leads to a break in schooling as well.

**Abortion as a synonym for complications.** Perspectives from community members also point to a consensus in the belief that abortion is synonymous with complications. When asked about their knowledge of complications and risks associated with abortion, the male respondents reported complications such as losing weight (thin and weak), psychological distress, death, while women cited infertility, infections, and subsequent miscarriages. Indeed, female respondents explained that women who have had one or several abortions would, in future, be unable to conceive due to damaging effects of abortion on their wombs. Others believe that any future pregnancies would always spontaneously terminate at the same gestational age of previous induced abortions. The quotes below illustrate this view about abortion:

*People would wonder whether she will ever conceive again. They will think because she had terminated her pregnancy, maybe she will not get a baby of her own. People say such things about them* (IDI, married woman, 33 years, Manyuanda, Rural setting).

*I have witnessed many girls who have procured abortions. They can't get children. They have become infertile. . . those who obtain an abortion within three months of pregnancy cannot carry any pregnancy beyond three months. Every time they are pregnant, they miscarry after every three months because the body was used to that during the days they were terminating the pregnancy* (FGD, Married women, 37 years, Manyatta, Urban setting).

The import of such community beliefs and perception is that young unmarried girls who have had or are perceived to have had an abortion are considered unsuitable for marriage. These women are regarded as failures linked to their inability to keep themselves "pure". In most communities where men (and their families) carry out informal background checks on women and their families before contracting marriages, women known to have had abortions are considered undesirable, loose, and not "wife material". Members of the community often assume that such women cannot have children and are "men" because they lost their womanhood (fertility) due to abortion. For some of these reasons, women may choose to relocate. However, those who remain in their community after procuring an abortion or come back after temporal relocation might be considered unsuitable for marriage. Young unmarried women in Nairobi and Kisumu felt that it is difficult to find a suitable marriage partner in the community if people are aware of one's abortion history.

*They are viewed as people who cannot be trusted. They cannot be trusted in marriage. That's what people say. Because she may not even want to take care of children in marriage* (IDI, Young Unmarried woman, 25 years, Viwandani, Urban setting).

In many communities in Kenya, children are a key indicator of marital success. Therefore, men are uncomfortable and unwilling to marry women who are known to have had one or multiple abortions due to the notion that such women may continue the practice. Interviews with a community health volunteer in an urban slums and a young woman from rural Kisumu demonstrate this opinion succinctly as captured in these excerpts:

*People are even warned against marrying those who have aborted... that they may continue with the abortion even in marriage. They speak evil things about them* (IDI, Community Health Volunteer, Jericho, urban setting).

*Yes, there is discrimination because they are not seen as other normal women. You might find a case where a man wants to marry a young lady but hesitates because he thinks the lady might abort his baby since she has done it before. So that's what happens* (IDI, Young woman, 23 years, Manyuanda, Rural setting).

**Normalizing abortion.**　For many of the young unmarried people (both females and males) interviewed, abortion is no longer a big deal, as women have a choice to make. Some reported that it is no longer shameful since it is carried out in secret, and there is no trail of evidence that someone procured an abortion unless there are health complications.

*Nowadays, it is not a big deal; it is a normal thing. It is like taking tea and bread* (FGD, Young unmarried women, 24 years, Viwandani, Urban setting).

*Abortion in this community is not a big deal. People will talk for about one week, and life continues* (FGD, Young unmarried men, 27 years, Viwandani, Urban setting).

Furthermore, many young men and women regard abortion as a private affair, especially if it is done during the early stages of pregnancy when it is not visually noticeable. Women felt their communities have evolved and the ease of accessing abortifacient products in most settings further protects women's privacy and ensures they obtain a safe procedure.

*There is no stigmatization because even though you know someone has aborted, you have no evidence because you did not witness. There is no witness or evidence for abortion* (IDI, Young unmarried woman, 26 years, Viwandani, Urban setting).

Participants cited "success stories" of women who procured abortion safely and whose lives were uninterrupted by complications:

*In the second abortion, she used mesofo* (misoprostol). *The lady told me that abortion by mesofo was safer compared to the other method she used. She said she didn't bleed as she bled in the first abortion. People could say that she will never conceive after procuring abortion twice. The lady recently conceived. So you can see that with mesofo, you can still get pregnant after abortion, and there aren't many side effects* (IDI, Female, 32 years, Manyatta, Urban setting).

## Awareness of, and access to medical abortion services

**Awareness of medical abortion products.**　Participants indicated awareness of the various ways through which women obtain abortion services. They cited a range of options, including women seeking safe abortion services in hospitals and pharmacies, to unsafe options from traditional birth attendants and self-induction. Nevertheless, a certain level of variability was

noted–women in urban areas (in both study counties) were aware of available medical abortion services in their community, whereas their counterparts in rural settings were mostly unaware of the availability of medical abortion, as the excerpts below demonstrate:

*I think most of them go to the quacks because they don't have the information about safer ways of securing an abortion* (IDI, Married male, 40 years, Gita, peri-urban setting).

*Even right now, I can tell you that we cannot go beyond four hundred meters if you need these drugs as long as you have at least 1000 or 1500 shillings* (FGD, Married men, 38 years, Viwandani, urban setting).

Lack of accurate and reliable information on the availability of safe medical abortion services within the community results in a substantial number of women seeking unsafe abortion services. Within rural areas, challenges go beyond just lack of information on abortion services, to inaccessibility of the drugs from many of the points of delivery, as the excerpts of an interview with a clinical officer in Gathiga, a peri-urban setting in the Nairobi outskirts, show:

*It is not very easy because even if you go to the chemist, and based on the regulations of selling drugs, you cannot be given the drugs unless it is a specific chemist- . . .So the drugs are not that accessible. For example, I cannot get the drugs from the chemists myself unless I am known there. If they don't know me, they will ask me for a prescription. If you don't have they will tell you they don't sell that one over the counter. They ask you to get a doctor's prescription. So they are not that accessible. That's why people are coming to the crude methods (*KII, Clinical Officer, 40 years, Gathiga, Peri-urban setting).

**Accessing medical abortion services.** Respondents recounted multiple, yet often complex pathways through which women and girls access medical abortion services. The pathways involved several tiers of middle-persons, including women who previously terminated pregnancies, community health volunteers and brokers. It was also clear that men in the slums of Nairobi were well aware of these networks:

*Here in this community, the doctors have certain people that bring them the clients, and they pay them a commission. . .so they have some middlemen* (FGD, Married men, 36 years, Viwandani, Urban setting).

*You cannot go to the chemist and openly say that you want to abort. Some women must connect you with the chemist. They are like the trustees of that chemist. You may never meet the doctor face to face. But the women act as the link, and they get the tablets for you. They are like brokers* (IDI, Young unmarried man, 27 years, Viwandani, Urban setting).

There was, however, limited knowledge and awareness of how exactly women and girls navigate the complexities of accessing medical abortion care. In almost all cases, respondents indicated that abortion is clandestine, and is often shrouded in secrecy as to who has sought the service, where and who delivers the service.

Pharmacies that stock abortifacients are often reluctant to sell directly to strangers (people they cannot verify as genuine clients), even when they have prescriptions–most likely due to the fear of repercussions as described by one pharmacy attendant in Kisumu County:

*Again the chemist cannot give the drug to someone they don't know because they know such a person may report them. I cannot just go to the chemists and buy the drugs. Even if I go with*

*the prescription they will not give me the drug. They only give it to someone they know* (IDI), Pharmacy attendant, 30 years, Manyatta, Urban setting).

There have been reports that pharmacy attendants express fears of arrests where police officers pose as clients, and therefore, the attendants always require proof of referral from a trusted source. In some instances, pharmacies might delay dispatching the drugs, requesting the women to come back at a later date to *"measure their seriousness"* or lack thereof. Women in an informal urban settlement describe it this way:

*Sometimes they think that you are on a mission to set them up. They become inquisitive and cannot directly sell the drug to you* (IDI, Married woman, 33 years, Viwandani, Urban setting).

*She had already visited the doctor who gave her the prescription, after which she sent me to town to buy the drugs for her. . . I had the prescription from the doctor. . . other chemists were refusing. So I went to different chemists. I walked into different chemists. I just looked for another chemist where I accessed the drugs* (IDI, Young woman, 21 years, Jericho, Urban setting).

Underscoring the complexity in accessing medical abortion drugs, women inherently end up delaying the timing of abortion, consequently elevating the risks of experiencing severe complications. Additionally, pharmacies operating within the communities are staffed by people who are familiar with women who might need abortifacients. Women, therefore, avoid pharmacies nearer to their communities, and resort to obtaining the abortifacient drugs from pharmacies that are several kilometers away from their usual places of residence to reduce the chances of recognition, as reported by a married man from an informal settlement in Nairobi:

*Yes, around here, I don't think you can even go to a counter and buy abortion medicine because as I have told you, most of these shops are run by our neighbors and relatives. So you cannot just go to your neighbor and tell them that you want to abort* (IDI, Married woman, 34 years, Jericho, Urban setting).

The fear of embarrassment is a compelling reason why women seek abortifacients in other communities, even when the abortion services are readily available in their locality. Laws surrounding abortion in Kenya are confusing and contradictory, fueling not only fears of victimization on the part of providers and suppliers of abortifacients, but also extreme secrecy among women seeking these services and the providers. These conditions, together, result in anxiety and desperation among women and young girls in need of pregnancy termination services, further subjecting them to exploitation by the pharmacy attendants, brokers and middle-persons, who, on the flip side, charge exorbitant prices for the drugs, partly due to the risks involved in the trade or just to profiteer off the women's desperation. This point is succinctly captured in the excerpts below:

*The pharmacist will hike the price because you are desperate. If he is a respected person, he will not refuse to sell the drug, but will only increase the amount* (FGD, Young male, 23 years, Manyatta, Urban setting).

*I can say that it is difficult; it is like they are hidden. That's why when someone has the products, they want to sell expensively because it is difficult to get. It is like they are hidden, and something hidden is often expensive, and since you are the one in need, the pharmacist will only do it for you if you can afford it* (IDI, Female, 40 years, Gita, peri-urban setting).

The cost of abortion drugs and services was ambiguous, and women have to negotiate with the service providers or suppliers. Young poor women with limited resources might not afford the high cost of safe abortion services. The majority of girls rely on boyfriends, friends, relatives, and mothers to raise money to access the services as reported by two of our respondents.

*. . .she does not have the money to purchase the drugs. So she will be forced to ask the boyfriend; mainly the women secure abortion for their daughters so that the husband does not find out* (IDI, Married male, 40 years, Gita, Peri-urban setting).

*The sponsors give them the money because their parents cannot provide them such amount of money every time. So if you are not working there must be another way. They are other boyfriends who accept responsibility for the pregnancy, but give you money to secure an abortion. That is the other way they get the money* (IDI, Young woman, 22 years, Jericho, Urban setting).

Similarly, women and girls in rural and peri-urban settings experience limited access to abortifacients and a general lack of awareness of the drug's availability, compared to women in urban settings.

*It is difficult to get them. Were it easy to get these medicines, these people who go to the quacks would not be going to the quack* (IDI, Married man, 27 years, Gita, Peri-urban setting).

Limited access and the high cost of safer abortion services inevitably lead to unsafe abortion practices and the resulting consequences.

## Discussion

Our study findings revealed that abortion is a common practice in the target communities, particularly among younger girls in urban settings, which is consistent with other studies [22, 23]. In a recent Guttmacher report on global abortion, researchers attributed the high abortion rates to the high unmet need for modern contraception, among the unmarried, sexually active women—especially adolescents, because stigma continues to impede them from getting contraceptive counseling and services [2]. Higher abortion rates have also been attributed to a combination of other factors, such as improved early detection of pregnancy and the wider availability of abortifacients such as Misoprostol drugs [24].

Our findings demonstrate that community perspectives around abortion and women who have sought such services are heterogeneous. The findings reflect a diversity of opinions ranging from those who are "totally against abortion" to the "neutral" and more "pro-choice" positions. Previous studies have shown that while initial community perspectives towards women who induced abortion are negative, these beliefs become more nuanced as details of the circumstances motivating women to seek abortion become apparent, such as terminating a pregnancy that resulted from rape [25], or when the abortion is performed in secret [26]. Our findings are in tandem with those of Casey et al. (2019) on community perceptions of induced abortion and post-abortion care in DRC. In both studies, respondents expressed multiple diverse opinions on abortion. Our findings are, however, more specific in the fact that the sympathy and compassion (which we characterized as "pro-choice") were more among young women in urban settings.

Additionally, as the findings show, women face a double challenge–having to move away from their usual residence to seek abortion services to avoid encountering people well-known to them and being denied those services by suspicious pharmacists. As already seen,

pharmacists are only comfortable selling to people they know and trust to be actual clients. It follows that unless something is done to change the abortion norms and context, women will continue to encounter multiple access-related challenges that impact their SRH.

Stigma around abortion remains dominant, and women and girls are ostracized and face isolation when discovered to have terminated a pregnancy. Yegon et al. concluded that abortion-related stigma plays a significant role in women's decision on whether to have a safe or unsafe abortion and that the incidence of unsafe abortion was higher in settings where abortion was more stigmatized [27]. The fear of social discrimination forces some women to relocate temporarily or permanently to new neighborhoods–causing severe disruption to their livelihoods and education for the girls. In addition, public knowledge that a girl has had an abortion appeared to reduce the girl's value and opportunities in the marriage "market" as Fergusson et al highlighted, while acknowledging that marriage is highly valuable in most societies, and defined and constructed as a crucial "rite of passage" into family life [28]. Consequently, social discrimination does not only affect women who have past abortion experiences, but also those who suffer fertility disorders occasioned by any other health or congenital reasons, or women who simply wish to limit or delay births. As a consequence of ostracism and isolation, the women and girls face risks of mental health disorders, self-harm, and suicide. Evidently, women who experience internalized shame and stigma after having an abortion experience increased psychological distress and physical health symptoms [29].

In spite of abortions being widespread, multiple barriers and complexities in accessing abortifacients in the study counties still exist, such as several middle-men and brokers who exploit desperate women and girls, unavailability, and high costs of drugs, and social stigma [16, 24]. There is a limited understanding of how women navigate these barriers while seeking abortion services. Paradoxically, these complexities and barriers are further exacerbated by the fact that providers were more comfortable offering abortifacients to women that were well-known to them versus women's desire to only seek these medications from places where they were unknown to the providers.

Unsafe abortions tend to thrive more in contexts where abortion laws are restrictive–both socially and legally [30, 31]. Women and girls are often dissuaded from seeking more overt abortions services, mostly out of fear of condemnation at public health facilities, uncertainty about the law, and perceived high cost of safe abortion methods. They, therefore, resort to cheaper and clandestine yet unsafe abortion services offered by untrained community midwives, drug sellers and/or in unequipped clinics [22]. In some cases, women decide to seek abortion services far away from their communities, or delay terminating pregnancies, as Marlo et al. report, where findings indicated that many people believed that abortion was safer at higher gestational ages [15]. Contrary to this belief, attempting abortion at advanced gestational age elevates the risk of complications and death exponentially.

This study's limitations include the possibility that respondents may have attempted to provide answers they consider as socially desirable, leading to a misrepresentation of their beliefs. Social desirability bias is more common in FGD settings, and this study mitigate this by combining both FGDs and IDIs. Even so, we believe this paper makes an important contribution to understanding the community perceptions and impediments to access to safe abortion services in Kenya.

## Conclusions

Our findings illustrate the perception of abortion being pervasive in Kenya, while women and girls contend with numerous and complex challenges to access safe abortion and quality post-abortion care. Also evident is the utter lack of skills and capability among girls to navigate the

complex barriers such as perceived legal restrictions, stigma, lack of awareness as well as culturally instigated gender vulnerabilities. Together, these factors not only deny women their inherent sexual and reproductive health and rights but also engender adverse sexual and reproductive health outcomes.

While Kenya is a signatory to key international and regional frameworks such as—the Maputo Protocol, which aims to enhance access to safe abortion [32], this aspiration remains far from reach. Thus, there is a compelling need to reform the Kenyan legal architecture to improve clarity as to the legality of abortion services. Ultimately, creating a more progressive and liberal environment for safe abortion service and strengthening the capacity of health providers to deliver services will lead to a reduction in unsafe abortions, complications related to unsafe abortions and ultimately, cause a significant reduction in maternal mortality in Kenya. Critically, efforts are needed to enhance education and awareness among 1) women to improve access to safe abortion, knowledge of strategic points of service access and understanding existing referral networks; 2) men by 'male/men-streaming' the abortion issue to raise awareness of their crucial role in the paths to unsafe abortion as a means of creating a more friendly environment for decision-making and safe care-seeking. Available evidence supports the role of community health volunteers in expanding access to safe abortion services. This study provides valuable evidence to inform interventions aimed at addressing the drivers of unsafe abortions in communities as well as ways of mitigating such gendered vulnerabilities, including the need for comprehensive sexuality education for young boys and girls.

## Author Contributions

**Conceptualization:** Boniface Ayanbekongshie Ushie, Kenneth Juma, Grace Kimemia, Michael Mutua.

**Data curation:** Kenneth Juma, Grace Kimemia.

**Formal analysis:** Kenneth Juma, Grace Kimemia.

**Project administration:** Kenneth Juma.

**Supervision:** Martin Bangha.

**Validation:** Ramatou Ouedraogo.

**Writing – original draft:** Boniface Ayanbekongshie Ushie, Kenneth Juma, Grace Kimemia, Michael Mutua.

**Writing – review & editing:** Boniface Ayanbekongshie Ushie, Kenneth Juma, Ramatou Ouedraogo, Martin Bangha, Michael Mutua.

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
