## [Decision Letter · Decision Letter 0]

10 Oct 2019

PONE-D-19-22660

Community perception towards abortion, women who abort and abortifacients in Kisumu and Nairobi Counties, Kenya

PLOS ONE

Dear Mr Juma,

Thank you for submitting your manuscript to PLOS ONE. After careful consideration, we feel that it has merit but does not fully meet PLOS ONE’s publication criteria as it currently stands. Therefore, we invite you to submit a revised version of the manuscript that addresses the points raised during the review process.

In your revisions, please address the reviewers' comments below.

We would appreciate receiving your revised manuscript by Nov 23 2019 11:59PM. To enhance the reproducibility of your results, we recommend that if applicable you deposit your laboratory protocols in protocols.io, where a protocol can be assigned its own identifier (DOI) such that it can be cited independently in the future. For instructions see: http://journals.plos.org/plosone/s/submission-guidelines#loc-laboratory-protocols

We look forward to receiving your revised manuscript.

Kind regards,

Denis Bourgeois

Academic Editor

PLOS ONE

**Journal Requirements:**

**Comments to the Author**

1. Is the manuscript technically sound, and do the data support the conclusions?

Reviewer #1: Yes

Reviewer #2: Yes

2. Has the statistical analysis been performed appropriately and rigorously? 

Reviewer #1: N/A

Reviewer #2: N/A

3. Have the authors made all data underlying the findings in their manuscript fully available?

Reviewer #1: Yes

Reviewer #2: Yes

4. Is the manuscript presented in an intelligible fashion and written in standard English?

Reviewer #1: Yes

Reviewer #2: Yes

5. Review Comments to the Author

Reviewer #1: The manuscript is original using cross-sectional exploratory study combining focus group discussions and in-depth interviews that explored community perceptions and beliefs relating to abortion, its users and abortifacients in Kenya. Both gender and healthcare providers (adult men and women, pharmacists, nurses, and community health volunteers) had participated in the study. The author noted that women who are seeking abortion are traumatized and excluded from society due to personal, religious, and social stigma.

The langage of the manuscript is correct and clear but we noted some specifics errors. The Introduction is lenghty, no need to provide a lot of information, revise it. Methods: The authors need to be more analytical, to provide statistical analysis by using weighting in the responses in order to make a hierachisation of the items. For a better understanding, use tables in the main Results. Be more precise in the Conclusion.

In the References, we note more specifics errors :

Reference 1: complete the pages number 130- ?

Reference 10: compléter the source

Reference 17: complete the pages numbers185- ?

Reference 24: re-format the title which is en capital

Reference 30 : provide journal name

Reference 33 : provide journal name

Reference 34: verify the page number there maybe too many figures

Reference 36: re-format the title which is in capital

Reference 37: complete this reference

This paper can be accepted once these modifications are made.

Reviewer #2: The work presented provides a strong case regarding the importance of addressing health policies regarding the provision of safe abortion services and post-abortion care from a multilevel perspective, from the necessary legislation, provider orientations, stock assurance and points of distribution, up to local interventions.

First, it contributes to the existing literature that links the religious and cultural values that build the stigma of abortion in the persistence of unsafe abortions in contexts where it is legally permitted. Secondly, and in our opinion its greatest contribution, makes visible the different barriers that women face to perform an abortion, which are of a material nature –lack of stock and distribution positions of the drug, as well as its high cost when operating criteria supply-demand–, but also socio-cultural –stigmatization, lack and/or inaccuracy of information on medical abortion, the complexity of the pharmaceutical-client relationship. The latter are key to the design of community interventions that reduce the gaps in access to safe abortion for women.

One of the innovative aspects of the study surveyed by the authors is the inclusion of men in the sample, in this sense, it would be interesting to discuss gender differences in perceptions regarding abortion. Although the results indicate the differences regarding the perception of abortion observed in rural and urban contexts, and at the gender and generation level (young non-urban women would be more open to understand abortion as a legitimate personal decision), it would be interesting to understand how this differentiation could be approached to positively impact the design of community interventions.

Finally, a revision of the text is recommended since some minor errors were found (eg discrepancy between the characterization of the appointment "married women" and the description thereof in the preceding paragraph "married men").

6. PLOS authors have the option to publish the peer review history of their article (what does this mean?). If published, this will include your full peer review and any attached files.

Reviewer #1: No

Reviewer #2: Yes: Claudia Moreno Standen

---

## [Author Response · Author response to Decision Letter 0]

14 Nov 2019

Response to reviewer’s comments

PONE-D-19-22660

Community perception towards abortion, women who abort and abortifacients in Kisumu and Nairobi Counties, Kenya

We appreciate the reviewers for the insightful comments and feedback on this manuscript. We agree with majority of the comments. In response, we have accordingly revised sections of the manuscript in line with the comments and where needed - provided additional details. We have included all changes in track and referenced the same in this letter. The references are captured in the revised manuscript with line references.

Reviewer #1: 

The manuscript is original using cross-sectional exploratory study combining focus group discussions and in-depth interviews that explored community perceptions and beliefs relating to abortion, its users and abortifacients in Kenya. Both gender and healthcare providers (adult men and women, pharmacists, nurses, and community health volunteers) had participated in the study. The author noted that women who are seeking abortion are traumatized and excluded from society due to personal, religious, and social stigma. 

The language of the manuscript is correct and clear but we noted some specifics errors. 

a. The Introduction is lengthy, no need to provide a lot of information, revise it. 

Response: We have substantially revised the introduction sections of the manuscript, and while attempting to provide greater basis for our study, we have tried to make the introduction more concise in the revise version (see page 3 and 4). 

b. Methods: The authors need to be more analytical, to provide statistical analysis by using weighting in the responses in order to make a hierarchisation of the items. For a better understanding, use tables in the main Results. 

Response: We thank the reviewer for this comment. We acknowledge the lack of weighting and hierarchization of the key themes, but would like to note that the aim of this study did not require a ranking of the perspectives emerging from the data. Moreover, we would like to emphasize that our study utilizes the conventional qualitative approach where we inductively identified emerging themes based on codes from interview transcripts, and then synthesized the information to outline key findings. We have also made the case, on page 5, line 147-158, by providing a detailed explanation of the analysis approach used, and how it is suitable for answering the research questions and overall aims.

c. Be more precise in the Conclusion.

Response: We have revised the concluding section of the manuscript to be more succinct, drawing a laser focus on the key take away from the study and precisely articulating the implications of such findings. These changes can be found in page 14 Line 470-485.

d. In the References, we note more specifics errors : 

i. Reference 1: complete the pages number 130- ?

ii. Reference 10: complete the source

iii. Reference 17: complete the pages numbers185- ?

iv. Reference 24: re-format the title which is en capital

v. Reference 30 : provide journal name

vi. Reference 33 : provide journal name

vii. Reference 34: verify the page number there maybe too many figures

viii. Reference 36: re-format the title which is in capital

ix. Reference 37: complete this reference

Response: We have updated the references accordingly. 

Reviewer #2: 

The work presented provides a strong case regarding the importance of addressing health policies regarding the provision of safe abortion services and post-abortion care from a multilevel perspective, from the necessary legislation, provider orientations, stock assurance and points of distribution, up to local interventions.

a. First, it contributes to the existing literature that links the religious and cultural values that build the stigma of abortion in the persistence of unsafe abortions in contexts where it is legally permitted. 

Response: We acknowledge the reviewer for this commendation and reiterate this in page 14 line 467.

b. Secondly, and in our opinion its greatest contribution, makes visible the different barriers that women face to perform an abortion, which are of a material nature –lack of stock and distribution positions of the drug, as well as its high cost when operating criteria supply-demand–, but also socio-cultural –stigmatization, lack and/or inaccuracy of information on medical abortion, the complexity of the pharmaceutical-client relationship. The latter are key to the design of community interventions that reduce the gaps in access to safe abortion for women. 

Response: Thanks for this commendation. We reiterated this in page 12 line 400; line 418-421; and page 13 line 438-445.

c. One of the innovative aspects of the study surveyed by the authors is the inclusion of men in the sample, in this sense, it would be interesting to discuss gender differences in perceptions regarding abortion. Although the results indicate the differences regarding the perception of abortion observed in rural and urban contexts, and at the gender and generation level (young non-urban women would be more open to understand abortion as a legitimate personal decision), it would be interesting to understand how this differentiation could be approached to positively impact the design of community interventions. 

Response: We appreciate this comment. We have added additional information capturing difference between men and women regarding abortion related complications on page 6, lines 203-208. We also provided insight on how interventions can build up on the gender difference in perceptions to target unsafe abortion drivers among women and men, page 14, lines 476-480.

d. Finally, a revision of the text is recommended since some minor errors were found (e.g. discrepancy between the characterization of the appointment "married women" and the description thereof in the preceding paragraph "married men"). 

Response: We appreciate this comment. We have reviewed the entire manuscript to correct errors pointed out, and those picked out by authors during this round of review. In doing this, we believe that we have improved the flow and accuracy of our report in the entire manuscript.

---

## [Editor Report · Decision Letter 1]

20 Nov 2019

Community perception of abortion, women who abort and abortifacients in Kisumu and Nairobi Counties, Kenya

PONE-D-19-22660R1

Dear Dr. Juma,

We are pleased to inform you that your manuscript has been judged scientifically suitable for publication and will be formally accepted for publication once it complies with all outstanding technical requirements.

With kind regards,

Denis Bourgeois

Academic Editor

PLOS ONE
---

## [Editor Report · Acceptance letter]

4 Dec 2019

PONE-D-19-22660R1 

Community perception of abortion, women who abort and abortifacients in Kisumu and Nairobi Counties, Kenya 

Dear Dr. Juma:

I am pleased to inform you that your manuscript has been deemed suitable for publication in PLOS ONE. Congratulations! Your manuscript is now with our production department. 

With kind regards,

on behalf of

Professor Denis Bourgeois 

Academic Editor

PLOS ONE